# Bacteria for Treatment: Microbiome in Bladder Cancer

**DOI:** 10.3390/biomedicines10081783

**Published:** 2022-07-25

**Authors:** Kyungchan Min, Hyun Tae Kim, Eun Hye Lee, Hansoo Park, Yun-Sok Ha

**Affiliations:** 1Department of Biomedical Science & Engineering, Gwangju Institute of Science and Technology, Gwangju 61005, Korea; minchance@gm.gist.ac.kr; 2Department of Urology, School of Medicine, Kyungpook National University, Kyungpook National University Hospital, Daegu 41944, Korea; urologistk@knu.ac.kr; 3Joint Institute for Regenerative Medicine, Kyungpook National University, Daegu 41940, Korea; eun90hye@gmail.com; 4Department of Urology, School of Medicine, Kyungpook National University, Kyungpook National University Chilgok Hospital, Daegu 41404, Korea

**Keywords:** microbiome, bladder cancer, *Bifidobacterium*, *Lactobacillus*, *Lactococcus*

## Abstract

The human body contains a variety of microbes. The distribution of microbes varies from organ to organ. Sequencing and bioinformatics techniques have revolutionized microbial research. Although previously considered to be sterile, the urinary bladder contains various microbes. Several studies have used urine and bladder tissues to reveal the microbiome of the urinary bladder. Lactic acid-producing bacteria, such as *Bifidobacterium*, *Lactobacillus*, and *Lactococcus*, are particularly beneficial for human health and are linked to bladder cancer. This review highlights the analysis protocols for microbiome research, the studies undertaken to date, and the microbes with therapeutic potential in bladder cancer.

## 1. Introduction

Bladder cancer (BCa) is one of the top ten most common urinary tract malignancies, with a 5-year mortality rate of 30% [1]. Based on the invasion of the muscular layer of the bladder, BCa can be divided into non-muscle-invasive bladder cancer (NMIBC) and muscle-invasive bladder cancer (MIBC). Approximately 80% of all BCa are NMIBCs, including low- and high-grade papillary neoplasms including carcinoma in situ (CIS) [2]. Several factors are associated with bladder carcinogenesis, and tobacco smoking is the most significant and most common factor [3]. Occupational exposure to carcinogens, such as aromatic amines (benzidine, 4-aminobiphenyl, 2-naphthylamine, 4-chloro-o-toluidine) and polycyclic aromatic hydrocarbons, contributes to approximately 20% of all bladder cancer cases [4,5,6]. Further, studies have suggested that genetic pathway mutations or alterations can contribute to bladder cancer. The tumor suppressor genes TP53, RB1, and PTEN are frequently mutated in CIS [7]. FGFR3, PIK3CA, and RAS are all oncogenes that promote tumor cell development and are characteristic of NMIBC [7,8].

Patients with NMIBC have relatively favorable outcomes, with a cancer-specific survival rate (CSS) of 70–85% at 10 years and a higher rate for low-grade disease [9,10]. NMIBC is a clinically heterogeneous group of cancers with a wide range of progression and recurrence probabilities. Approximately 55% of low-grade Ta lesions recur after a long-term follow-up, but only 6% progress to stage 3. In contrast, high-grade T1 lesions are at significant risk of recurrence (45%) and progression (17%) [11]. An analysis of several prospective Radiation Therapy Oncology Group (RTOG) protocols investigating bladder-preserving combined-modality therapy for MIBC with a median follow-up of 4.3 years found that the overall 5- and 10-year survival rates were 57% and 36%, and the disease-specific 5- and 10-year survival rates were 71% and 65%, respectively [12].

According to the National Comprehensive Cancer Network (NCCN) guidelines, transurethral resection of bladder tumors (TURBT) and postoperative chemotherapy are sufficient in a patient with low-risk NMIBC [13]. However, urologists should consider intravesical therapy if the tumor progresses to intermediate or high risk. Bacillus Calmette–Guérin (BCG) or radical cystectomy should be considered for high-risk, BCG-naive patients. If the patient is intolerant or non-responsive to BCG, clinicians should consider cystectomy, intravesical chemotherapy, or pembrolizumab [13]. A key question in NMIBC research is how to predict in advance which patients will respond to BCG immunotherapy, or which patients will progress or recur into a muscle-invasive phenotype. In addition, BCa recurrence after BCG therapy for the primary tumor remains a challenge [14].

Many treatments can be considered for patients with MIBC. Depending on the patient’s condition, radical cystectomy with or without systemic chemotherapy, concurrent chemoradiotherapy, or radiotherapy alone may be considered in localized disease [13]. Chemotherapy and immune checkpoint inhibitors (ICI) can be used if the cancer is metastatic [13].

Considering the significant loss of quality of life by repeated TURBT and radical cystectomy in NMIBC, several studies have been conducted to prevent progression and recurrence, and to preserve the bladder. Immune checkpoint inhibitors have proven to be a successful therapeutic strategy for treating metastatic bladder cancer. For urothelial cell carcinoma of the bladder, PD-L1 inhibitors such as avelumab and atezolizumab, and PD-1 inhibitors such as nivolumab, pembrolizumab, and durvalumab have been approved. Several clinical trials are currently underway, in which the drug combination is tested in both neoadjuvant and adjuvant settings [15]. In particular, nadofaragene firadenovec and enfortumab vedotin are novel drugs. In total, 151 patients with NMIBC who did not respond to BCG were treated with intravesical rAd-IFNα/Syn3 in a phase 3 open-label trial of nadofaragene firadenovec [16]. Among 103 patients with CIS, 55 (53.4%) remained responsive for three months after the first dose, and 25 (45.5%) remained free of recurrence for 12 months following the first dose. In the global, open-label, phase 3 trial of enfortumab vedotin, 301 patients with locally advanced or metastatic urothelial carcinoma, who had previously received platinum-based treatment and a PD-1 inhibitor or PD-L1 inhibitor, were enrolled [17]. The antibody–drug conjugate (ADC) enfortumab vedotin targets nectin-4, a cell-adhesion molecule expressed in urothelial cancers. The enfortumab vedotin group showed longer overall survival and progression-free survival than that of the conventional chemotherapy group. The immune system has been shown to influence response to BCG therapy [18], and immunological research serves as a driving force for the formulation of novel paradigms to treat BCa.

The microbiome is the collection of microbes, their products, and their surrounding environment [19]. Using bioinformatics and sequencing technologies, many different microbes have been found to live in various sites of the human body, including the urinary tract, which is commonly believed to be sterile [20,21]. For microbiome-based therapeutics, fecal microbiota transplantation, diet and prebiotics, symbiotic microbiota consortia, engineered symbiotic bacteria, and microbiota-derived proteins and metabolites have been studied [22]. Moreover, some metabolites, including short-chain fatty acids, are involved in the mechanisms of the microbiome [23]. Increasing evidence indicates that the microbiome influences the development and treatment of genitourinary malignancies [24,25,26,27].

As therapeutics in recurrent NMIBC are related to immunity, as stated above, the microbiome has the potential to be used for new treatments beyond BCG. An understanding of how the bladder microbiota affects immunotherapy response, leading to patients who are non-responsive to BCG treatment, is necessary [28]. Researchers are gaining insights into how the bladder microbiome affects therapy response with new NGS techniques. BCG treatment-related high recurrences and progressions may be explained by microbiome composition [14]. Subsequent studies have indicated that the microbiome of bladder cancer involves NK and T cell activities, suggesting relevance to immunity. 

This review summarizes the methods for studying the microbiome, the studies undertaken to date, and the beneficial microbes in bladder cancer.

## 2. Methods for Microbiome Study in Bladder Cancer

Researchers generally use urine samples and tissues to study bladder cancer. Despite stools being the most widely used material for other types of studies, urine and tissues are used the most for bladder cancer because of the location of the disease [29,30,31,32,33,34,35,36,37,38,39,40,41,42,43]. 

For microbiome research, researchers should first extract DNA from the sample materials. Researchers can easily extract DNA from stools as they contain a large amount of DNA. However, urine has a low DNA content, and it is difficult to extract DNA from urine. It is therefore advisable to use DNA extraction kits tailored for urine. Several commercial kits are available for DNA extraction from urine [44]. In addition, one should be aware of contamination and determine the appropriate methods for collecting urine. According to Hourigan et al. [34], the urinary microbiome varies with the collection methods used and sex. They compared urine obtained at midstream with that obtained through cystoscopy, and found that the method of collection and patient sex can influence differences in microbial abundance, indicating the importance of the collection method. This was confirmed by Oresta et al. [42]. The midstream and catheterized urines showed differential taxa abundances, and washouts of bladder cancer showed a different abundance from that of catheterized urine samples without washout, even though there were no differences in species diversity. To ensure the appropriate comparison between groups, they recommend that sampling methods need to be controlled.

After DNA extraction, researchers should decide whether to perform 16S rRNA gene amplicon sequencing or shotgun metagenomic sequencing. 16S rRNA gene amplicon sequencing is a type of amplicon sequencing using only partial variable regions that are discriminative between microbes [45]. Shotgun metagenomic sequencing uses short fragmented reads from whole microbial genomes derived from the patient samples [46]. Although shotgun metagenomic sequencing provides more information than 16S rRNA sequencing, 16S rRNA sequencing remains popular because of its cost-effectiveness. In cases where a researcher is trying to identify species or even strains from DNA, shotgun metagenomic sequencing is a better choice. Following the selection of methods, barcoding/adapter ligation, amplification, library preparation, and sequencing are carried out using Illumina or BGI devices. These processes are generally performed by qualified organizations or companies after DNA extraction [47].

These experimental processes must be followed by computational processes (bioinformatics analyses) to produce readable results. These processes on Linux include quality control, trimming, clustering, assignment to the reference database, functional classification, and others. Cutadapt, Trimmomatic, and fastp are the common tools used for quality control [48,49,50]. Although adapter and primer sequences need to be specified in cutadapt, Trimmomatic and fastp are automated tools that do not require the specification of sequences. Following quality control, Mothur and QIIME2 may be used in subsequent 16S rRNA sequencing analyses [51,52]. The Mothur application is programmed in C, and QIIME2 is written in Python. Currently, QIIME2 is the most popular application owing to its user-friendly interface. QIIME includes some processes that can be performed with just a single click on the website (https://qiime2.org (accessed on 10 June 2022)). Further, QIIME2 is useful if the user wishes to conduct functional analyses in addition to the taxonomic assignment. The most widely used functional analysis tool, PICRUSt2, is easily compliant with QIIME2 [53].

Kraken2 and MetaPhlAn3 are useful tools for shotgun metagenomic sequencing [54,55]. Metagenomic compositional profiling can be achieved either through the precise alignment of k-mers with a classification algorithm (Kraken) or by identifying unique clade-specific marker genes from three thousand reference genomes (MetaPhlAn). Researchers can use these tools to investigate microbial diversity and abundance. HUMAnN3, MEGAN, and other platforms for shotgun metagenomic sequencing can be used to conduct functional analyses [55,56]. HUMAnN uses a biphasic alignment screen with MetaPhlAn, followed by functionally annotated pangenomes of the identified species. These processes have been well described elsewhere [47]. Computational methods are not strictly defined, and many different combinations are possible at each step. Accordingly, users generally set up and customize their own pipelines.

## 3. Microbiome Research in Bladder Cancer

Urine and tissues are generally used for studies of the microbiome of bladder cancer, as discussed in Section 2. In some studies, both are used. In this review, we emphasize beneficial microbes at the genus and species level, as the review is about bacteria for treatment. This section is summarized in Table 1.

### 3.1. Research Using Urine

Research on the bladder microbiome began with an analysis of urine samples. To date, all studies have utilized 16S rRNA sequencing. In 2018, two studies analyzed urine samples from patients with bladder cancer. Popovic et al. [29] evaluated urine from 12 patients with bladder cancer and 11 healthy, age-matched controls. Samples were collected from people without prior antibiotic administration. No significant differences in diversity or abundance were found between the two groups. Wu et al. [30] collected midstream urine samples from 31 male patients and 18 non-neoplastic controls. Observed species, Chao1, and Ace indexes showed greater alpha diversity in the patients with cancer. They found that some taxa were significantly different at the genus level based on malignancy, risk of recurrence, and progression. They also used PICRUSt and found that *Staphylococcus aureus* infection, glycerolipid metabolism, and retinol metabolism pathways were enriched in patients with cancer. Despite the improvement in methodology, no further study was conducted on metabolic pathways to identify the meaning of these observations.

In 2019, Bi et al. [31] collected urine samples from 29 patients with cancer and 26 patients without cancer. One of the characteristics of this study is that caution was taken to avoid contamination. Researchers used only the taxa present in more than 1% of the samples for bioinformatics analyses. The groups showed different levels of diversity and abundance at the genus and species levels. Patients with cancer showed increased alpha diversity in the aspects of Chao1. Notably, probiotics *Bifidobacterium* and *Lactobacillus* were abundant in the control group. Additionally, they suggested *Actinomyces europaeus* as an indicator of bladder cancer.

In 2020, Chipollini et al. [33] collected midstream urine samples from 15 patients with muscle-invasive BCa, 10 with non-muscle-invasive BCa, and 10 controls. Compared with Bi et al., they reported different results [31], showing decreased alpha diversity among malignant groups and no difference in beta diversity. Further, there was no difference in microbial abundance. Zeng et al. [38] collected midstream urine samples from 40 patients with NMIBC and 19 healthy controls. Researchers used the International Prostate Symptom Score (IPSS) to exclude patients with severe lower urinary tract symptoms (LUTS). A higher level of alpha diversity was found in patients with malignancy in the aspects of Chao1 and Ace indexed compared to that in the controls, supporting the reports of Wu et al. [30] and Bi et al. [31], but differing from that of Chipollini et al. [33]. In terms of both the Shannon and Simpson indexes, patients with recurrence showed greater alpha diversity. The low Shannon index was associated with a higher recurrence-free survival rate (*p* = 0.019). Additionally, they found that *Lactobacillus*, which is generally beneficial in other diseases, was more abundant in the non-recurrent group (*p* = 0.06). This is consistent with the result of the study by Bi et al. [31].

Hussein et al. [39] collected midstream and catheterized urine samples from 43 patients and 10 controls in 2021. A comparison was made between cancer vs. healthy cohorts, NMIBC vs. MIBC, BCG responders vs. non-responders, and male vs. female patients. Notably, *Lactobacillus* was more abundant in healthy controls than in patients with cancer, in agreement with findings from Bi et al. [31] and Zeng et al. [38]. The other parameters also showed differences, but beta diversity was the only factor indicating a difference between cancer and healthy cohorts. Ma et al. [41] focused on smoking. The researchers collected urine samples from 11 healthy controls and 15 patients with bladder cancer—the cohorts comprised non-smokers and smokers. Comparing the groups revealed different biodiversity and different taxon abundances, which even varied between species and different functional pathways. Notably, *Lactobacillus* species and genera were enriched in non-smokers. Furthermore, they suggested that smoking impacts bladder cancer incidence through changes in the urinary microbiome [41].

In summary, the microbial abundance in patients with bladder cancer certainly differs from that in the controls. Catheterized urine is also different from voided urine, as discussed in Section 2 [34,42]. There is, however, no typical result for taxa abundance across studies, possibly because of various factors, including the small sample size, collection methods, and differences between hospitals. To identify the urinary microbiome in bladder cancer, additional research needs to be conducted.

### 3.2. Research Using Tissue

Unlike studies using urine, diverse sequencing methods have been employed in studies using tissues. According to Liu et al. [32], 22 carcinoma tissues and 12 adjacent normal tissues were collected. During the surgical procedure, bladder carcinoma tissue samples were taken from the mucosal surface of the bladder, and normal tissues were located approximately five centimeters away from the cancerous tissues. Participants were carefully selected based on their medication history and comorbidities. Using 16S rRNA sequencing, researchers discovered lower alpha diversities in the aspect of the Shannon index in cancerous tissues. *Lactobacillus* was found to be more abundant in other normal tissues, as shown in studies using urine [31,38,39,41].

Rodriguez et al. and Li et al. took different approaches. Rodriguez et al. [37] performed microbial detection using whole-exome sequencing (WES) data from multiple cancers in the Cancer Genome Atlas (TCGA) database. By using the Pathoscope 2.0 program, they identified enriched populations of *Cutibacterium acnes*, *Mathylobacterium radiotolerans*, *Pseudomonas aeruginosa*, and *Pseudomonas putida* in bladder cancer tissues. It is unique for this study to use WES data in contrast with previous microbiome studies. Furthermore, Li et al. [40] utilized whole transcriptome sequencing (WTS) data from TCGA to study the microbiome. Pathoscope 2.0 was used for contamination screening and microbial read extraction. Furthermore, the research focused on the relationship between the microbiome and epithelial-to-mesenchymal transition (EMT). The researchers found that *Escherichia coli* K-12, *Saccharomonospora viridis*, *Escherichia coli* O157: H7, Butyrate-producing bacteria SM4/1, and *Oscillatoria* sp. CCAP 1469/13 showed a significant correlation with genes associated with EMT/fibrosis. Additionally, they found that *Citrobacter koseri*, *Cupriavidus taiwanensis*, *Escherichia coli* O157: H7, *Bradyrhizobium denitrificans*, Flavobacteriaceae bacteria, and other species had a significant relationship with genes involved in the extracellular matrix (ECM) and with clinical variables such as race, tumor stage, and pathology. This study is remarkable in its application of WTS data for studying the microbiome and the computational decontamination method.

Chen et al. [43] conducted another trial. The authors sought to determine the relationship between PD-L1 expression and microbial abundance. They collected tissues from patients with NMIBC positive for PD-L1 on IHC (*n* = 9) and negative for PD-L1 on IHC (*n* = 19). In the PD-L1 group, alpha diversity was higher in both the Ace index and observed species; *Leptotrichia* was more abundant, and *Prevotella* was less abundant.

Dohlman et al. [57] recently published The Cancer Microbiome Atlas (TCMA) using WES and WGS data from 3689 gastrointestinal cancer samples from TCGA. They performed acquisition, metagenomic profiling, and decontamination of TCGA sequencing data using their computational pipelines, as well as testing and validation in vitro. To perform a microbiome study using TCGA data, a complex decontamination process using computational tools must be applied. Although the atlas only covers oropharyngeal, esophageal, gastrointestinal, and colorectal tissues, researchers unfamiliar with the complex computational process may find that using the TCMA database is extremely helpful. The data can be accessed at https://tcma.pratt.duke.edu (accessed on 10 June 2022).

In summary, few studies have been conducted on tissue analysis. Contrary to studies using urine samples, data from 16S rRNA sequencing, whole-exome sequencing, and whole transcriptome sequencing were used in the tissue study. Although TCGA datasets are not particularly designed for microbiome studies, researchers can use TCMA to obtain standardized microbiome data from TCGA. Overall, more research is needed for microbiome studies in tissues.

### 3.3. Research Using Both Urine and Tissue

Researchers conducted two studies using urine and tissue. These studies aimed to determine the relevance of the two samples. Mansour et al. [35] collected 10 catheterized urine samples and 14 malignant mucosal tissue samples removed during TURBT, matched from 10 patients with bladder cancer. The analyses were carried out using 16S rRNA sequencing and the Kraken pipeline. Alpha diversities in the Shannon index between males and females in the tissue microbiome were significantly different. Different tissues from the same patient showed virtually the same microbial composition. Moreover, there were differences in microbial abundances based on age group, gender, and type of sample. For example, *Akkermansia*, *Bacteroides*, *Clostridium*, *Enterobacter*, and *Klebsiella* were more abundant in tissues than in urines. This study is remarkable in showing the relevance of the urine microbiome to the tissue microbiome at the genus level. In this study, many microbial genera such as *Staphylococcus* and *Lactobacillus* were found in both urine and tissue samples.

Pederzoli et al. [36] conducted a similar study but with a larger sample size. In 21 men and eight women, midstream urine and non-tumorous tissues were matched. Further, midstream urine samples were collected from 15 male patients, 5 female patients, 34 healthy men, and 25 healthy women. Sequencing of 16S rRNA was performed along with gender discriminating analyses. Significant differences in microbial abundance were found between the urine samples of male patients and healthy controls. The urine samples of female patients and healthy controls also showed significant differences. Regardless of gender, malignant tissues contained more *Burkholderia* than non-malignant tissues. Additionally, this study indicated that more than 80% of bacterial families were shared between urine and tissue (malignant and non-malignant) microbiomes. However, the level was restricted to a family, which is too large to discriminate.

Ultimately, urine and bladder tissue share a high proportion of the microbiome. Owing to the limitations of sequencing and computational tools, the level of microbial discrimination is not sufficiently deep. As shotgun metagenomic sequencing contains more information than 16S rRNA sequencing, studies using such advanced techniques might identify the microbes more clearly.

## 4. Beneficial Microbes in Bladder Cancer

Lactic acid is a well-known metabolite for its advantages to human health. Lactic acid-producing bacteria can be divided into two categories: Lactic acid bacteria (LAB) and *Bifidobacterium*. *Lactobacillus*, *Lactococcus*, *Streptococcus*, etc., are among the commonly known LAB [58]. Fermented products are known to contain LAB. Several cohort studies have been conducted on fermented products rather than on specific species.

Ohashi et al. [59] conducted a case-control study including 180 cases and 445 controls. In this study, the cases were patients with recently diagnosed bladder cancer, and the controls were healthy individuals without bladder cancer. In questionnaires, researchers asked about the consumption of fermented milk products or Yakult for a period of 10–15 years. Smoking and consumption of fermented milk products and Yakult were significant in conditional logistic regression analysis. Low intake of fermented milk products and Yakult was associated with an increased incidence of bladder cancer. Further, extensive cohort studies commonly concluded that fermented milk products such as yogurt or sour milk were significantly associated with bladder cancer incidence in Cox proportional hazard regression analyses [60,61,62]. Table 2 summarizes the following.

### 4.1. Bifidobacterium Species

Below are some studies on *Bifidobacterium infantis* (BI)-mediated herpes simplex virus thymidine kinase/ganciclovir (HSV-TK/GCV) suicide gene therapy. Cinque et al. [63] have demonstrated that BI can induce keratinocyte apoptosis in response to activated T lymphocytes. Tang et al. [64] hypothesized that this characteristic of BI could be extended to anaerobic tumors, and conceptualized a suicide gene therapy system with the premise that thymidine kinase expressed in tumor tissues can convert ganciclovir into a toxic substance that kills tumor cells. To generate pGEX-TK, the HSV-TK gene was subcloned into pGEX-5X-1. Furthermore, pGEX-TK was electroporated into BI bacterial cells. In a bladder cancer rat model induced by N-methyl-nitrosourea (MNU), they found that BI-mediated TK/GCV suicide could effectively inhibit tumor growth by increasing caspase-3 expression and inducing apoptosis.

Additionally, Yin et al. [65] showed that treatment with BI-TK/GCV increased the expression of Fas, FasL, Cyt-C, and Caspase-9, suggesting that BI-TK/GCV acts through both extrinsic and intrinsic pathways of apoptosis. Jiang et al. [66,85] conducted proteomic analyses of the system following the study. The cancer proliferation-associated protein PCNA, glycolysis-associated protein PKM2, HKK-1, and PFK-B, and invasion-associated protein CD146 were found to be downregulated in tumor tissues of BI-TK/GCV treated rats [85]. They discovered that Prx-I expression was decreased in the tumor tissues of rats treated with BI-TK/GCV. Furthermore, they found that this downregulation suppressed tumor growth, improved apoptosis, and was associated with the NF-kB pathway [66].

Research has been conducted on developing a vaccine for oral cancer using a Bifidobacterium longum (BL) vector. Kitagawa et al. [67] examined the possibility of combining oral cancer vaccines to overcome resistance to immune checkpoint immunotherapy. Furthermore, they demonstrated that an oral cancer vaccine combined with anti-PD-1 antibodies is efficacious in mouse models of bladder cancer. They developed a recombinant BL that contains Wilms’ tumor 1 (WT1) protein as a cancer vaccine. With the help of bladder cancer mouse models injected with MBT-2 cells, which naturally express WT1 proteins, these researchers found that the vaccine can boost the antitumor activity of anti-PD-L1, and even accomplish this through the expansion of CD4 and CD8 T cells alone.

### 4.2. Lactococcus Species

The effects of *Lactococcus lactis* on bladder cancer have been studied. Asensi et al. [68] created *Lactococcus lactis* (LL) secreting attenuated recombinant staphylococcal enterotoxin B for oral administration. In a model of *Staphylococcus aureus* infection, oral immunization with LL-rSEB induced a protective immune response. Using this study as a basis, Reis et al. [69] used an NMU-induced rat bladder cancer model to demonstrate that LL-rSEB is more effective than BCG therapy. Through intravesical injection of LL-rSEB, a better balance was observed between apoptosis and cell proliferation in bladder tissues with decreased VEGF, HIF, NOX4, and MMP-9 levels. This study is notable for showing that *Lactococcus lactis* is related to the treatment of bladder cancer, similar to *Bifidobacterium infantis* and *B. longum*.

### 4.3. Lactobacillus Species

There have been significant studies on *Lactobacillus casei* strain Shirota (LcS) and *Lactobacillus rhamnosus* strain GG (LGG) since the 1980s and 2000s. As these two species share many characteristics, Seow et al. [86] conducted in vitro studies using these two strains and cell lines. Their studies demonstrated that *Lactobacillus* species exerted anti-proliferative effects on bladder cancer cell lines. Sections four and five summarize the studies conducted on each species.

### 4.4. Lactobacillus Casei

As LcS has been sold as a commercial product, Yakult, since 1935, the majority of research using this strain has been conducted in Japan. Asano et al. began this research in 1986 [70]. They found that LcS feeding and intravenous injection effectively inhibited tumor growth using a subcutaneous MBT-2 implanted mouse model. Aso et al. [71] conducted a double-blind trial with oral intake of LcS in 138 patients with superficial bladder cancer. Patients with multiple primary or recurrent single tumors, other than multiple tumors, experienced longer recurrence-free survival following LcS consumption. Additionally, they found a very low incidence of adverse events. Naito et al. [72] conducted a randomized controlled trial. Compared with the intravenous administration of epirubicin, oral administration of LcS showed a higher rate of remission at three years and a similar rate of adverse events among patients with superficial bladder cancer. Therefore, they suggested that oral intake of LcS could be effective in preventing recurrence.

Further studies were conducted to elucidate the mechanism in addition to these cohort studies. Matsuzaki [73] used B16 and line-10 hepatoma tumor cell lines in mouse and guinea pig models. The researcher demonstrated anti-metastatic effects with intravenous and intralesional injections of LcS through NK cell activation. Subcutaneous injection of LcS before Lewis lung carcinoma inoculation (priming) resulted in increased CD4 T cell populations in peritoneal exudate cells. Additionally, this model showed an increase in IL-1β, IL-2, TNF-α, and IFN-γ in the peritoneal cavity, suggesting that these cytokines were associated with the anti-metastatic effects of LcS. Moreover, the researcher implanted a Meth A tumor intrapleurally into BALB/c mice. Intrapleural injection of LcS improved survival in this model and increased the levels of IFN-γ, IL-1β, TNF-α, NOx, IL-4, IL-6, and IL-10 in the thoracic cavity. 

Kato et al. [74] conducted in vitro and in vivo studies. An in vitro study revealed that LcS induced the release of IL-12 and IFN-γ from mouse splenocytes, likely promoting macrophage activation. In vivo experiments showed increased production of IFN-γ in mice spleens. Takahashi et al. [75] carried out a similar in vitro study with additional intravenous injection of LcS in subcutaneous and orthotopically implanted MBT-2 mice. In both in vivo models, LcS subcutaneous and intravenous injections resulted in higher tumor regression rates, as well as reduced tumor weights, showing antitumor effects. According to the orthotopic model, intravesical injection of LcS results in higher levels of IFN-γ and TNF-α transcription in bladder tissues. All three studies showed the involvement of T lymphocytes and IFN-γ in LcS-induced mechanisms.

Research concerning the mechanism of LcS in NK cells was conducted after these studies. Hori et al. [76] administered oral doses of LcS to aged mice, which are known to have low NK cell activity, to examine whether this substance affects NK cell function. Upon LcS feeding, they observed increased NK cell activity in blood mononuclear cells and splenocytes from aged mice. Moreover, Shida et al. [77] demonstrated that monocytes are essential for IFN-γ-secreting T cell activity and NK cell activity in the model of LcS coculture with PBMCs from healthy people.

Then, several studies were conducted in human cohorts. According to Takeda et al. [78], peripheral blood NK cells were significantly increased in middle-aged subjects (30–45 years) with low NK cell activity before LcS intake. Dong et al. [79] found that elderly healthy volunteers (55–74 years) with oral LcS intake showed higher proportions of monocytes, CD8- NK cells, and helper T cells. Additionally, the group showed a decrease in granulocytes, cytotoxic T cells, and CD8+ NK cells. The markers CD25 and CD69 for lymphocyte activation and concanavalin A (ConA) for T cell stimulation were used for detailed analyses. Increased CD69+ helper T cells and CD25+ helper T cells were observed in the LcS intake group. Furthermore, the ratio of IL-10 to IL-12 in the elderly also demonstrated a significant difference, indicating an anti-inflammatory response to oral LcS intake. On the contrary, a study involving healthy males aged 18–60 years showed no difference in NK cell activity following oral administration of LcS [80].

### 4.5. Lactobacillus rhamnosus

*Lactobacillus rhamnosus* strain GG (LGG) is an extensively studied microbe. Lim et al. [81] performed an animal experiment using subcutaneously implanted MB49 cells in mice. LGG feeding was found to decrease tumor size, increase T lymphocyte populations in the spleen, and increase lymphocyte populations in the tumors. Although they used a subcutaneously implanted model, this study demonstrated a significant relationship between LGG and tumor regression.

Two studies were published by Seow et al. regarding the effectiveness of LGG in the mouse bladder. They showed that intravesical LGG injections increased some genes associated with immune function and NK cell populations in healthy mouse bladders and in iliac lymph nodes, similar to that with BCG [82]. Following the study, they injected LGG and BCG intravesically into orthotopic mouse models. They showed that

(a)As with BCG, LGG intravesical injection induced tumor regression.(b)Intravesical LGG injection along with LGG feeding led to similar results to an intravesical injection alone.(c)Similar to lyophilized LGG, live LGG showed a slightly different effect.(d)Protein analyses indicated that LGG therapy could inhibit enzymes associated with metastasis and tissue remodeling.(e)Infiltration of macrophages and neutrophils into the tumor was induced by LGG [83].

To understand the mechanism of LGG, Cai et al. [84] conducted in vitro experiments using mouse-derived neutrophils, dendritic cells, and T lymphocytes. They found that IFN-γ and IL-2 secretion by T cells were related to LGG dose and exposure time. Activation of dendritic cells and neutrophils played a major role in this process.

## 5. Concluding Remarks and Future Perspectives

Bioinformatics analyses using 16S rRNA sequencing have been widely used for microbiome analyses in bladder cancer. Studies have been conducted to demonstrate the efficacy and mechanism of action of some lactic acid-producing bacteria. This review explains that beneficial microbes associated with bladder cancer are associated with immune-related mechanisms. Owing to their low toxicity and high efficacy, microbes have the potential to replace and supply ICI. This area of research has significant potential, as many types of bacteria are yet to be studied in bladder cancer. Multi-omics research involving metabolomics can help identify mechanisms of action. Furthermore, microbial signatures can be developed for various purposes, such as predicting the response to therapeutics or correlating the microbiome and genomic signatures.

## Figures and Tables

**Table 1 biomedicines-10-01783-t001:** Summary of microbiome research in bladder cancer.

Material	Author	Cohorts and Diversity	Abundance and Other Findings
Urine	Popovic et al. [29]	- Cancer vs. healthy - No significant difference between groups	- No significant difference between cancer and healthy group
	Wu et al. [30]	- Cancer vs. healthy - Increased alpha diversities in cancer group	- More abundant *Aeromonas, Acinetobacter, Bacteroides* in cancer group - More abundant *Proteus, Laceyella, Serratia* in healthy group
	Bi et al. [31]	- Cancer vs. healthy - Increased alpha diversities in cancer group	- More abundant *Streptococcus, Bifidobacterium, Lactobacillus, Veillonella* in healthy group - More abundant *Actinomyces europaeus* in cancer group
	Chipollini et al. [33]	- Cancer vs. healthy - Decreased alpha diversities in cancer group	- No significant difference between cancer and healthy group
	Zeng et al. [38]	- Cancer vs. healthy - Recurrent vs. nonrecurrent cancer - Increased alpha diversities in cancer group compared to healthy group - Increased alpha diversities in recurrent group compared to nonrecurrent group	- No significant difference between cancer and healthy group - More abundant *Lactobacillus* in nonrecurrent group compared to recurrent group
	Hussein et al. [39]	- Cancer vs. healthy - Different beta diversities between cancer and healthy group	- More abundant *Lactobacillus* in healthy cohort
	Ma et al. [41]	- Cancer vs. healthy - Nonsmoker vs. smoker - Decreased alpha diversities in smokers with cancer compared to nonsmokers with cancer	- More abundant *Lactobacillus* in nonsmoking patients
	Hourigan et al. [34]	- Midstream urine vs. cystoscopy - Male vs. female - No significant difference according to urine collection methods and gender	- *Stenotrophomonas* increased in cystocopy - *Tepidomonas* increased in males - *Prevotella* and *Veillonella* increased in females
	Oresta et al. [42]	- Cancer vs. healthy - Midstream urine vs. catheterized urine - Catheterized urine vs. washout urine - Increased alpha diversities in cancer patients	- *Veillonella*, *Corynebacterium* increased in cancer patients - *Ruminococcus*, Enterobacteriaceae increased in healthy controls - *Streptococcus*, *Enterococcus*, *Corynebacterium*, *Fusobacterium* increased in midstream urine - Ruminococcaceae decreased in midstream urine - Buckholderiaceae, *Faeclibacterium*, *Erysipelatoclostridium*, *Veillonella*, *Streptococcus* differ between catheterized and washout urines
Tissue	Li et al. [32]	- Carcinoma vs. adjacent normal tissue - Lower alpha diversities in cancerous tissues	- More abundant *Lactobacillus*, *Prevotella*, Ruminococcaceae in normal tissue - More abundant *Cupriavidus*, *Acinetobacter*, *Anoxybacillus*, *Escherichia-Shigella*, *Geobacillus, Pelomonas, Ralstonia, Sphingomonas* in cancerous tissue
	Rodriguez et al. [37]	- Carcinoma vs. adjacent normal tissue (TCGA, WES)	- More abundant *Cutibacterium acnes, Mathylobacterium radiotolerans, Pseudomonas aeruginosa* and *Pseudomonas putida* in bladder cancer tissues
	Li et al. [40]	- MIBC tissue only (TCGA, WTS)	- Relevance between microbial abundances, EMT/fibrosis/ECM-related genes, and clinical variables
	Chen et al. [43]	- NMIBC tissue with PD-L1 (+) vs. PD-L1(-) - Increased alpha diversities in PD-L1 (+) tissues	- More abundant *Leptotrichia* and less abundant *Prevotella* in PD-L1 (+) tissues
Urine and Tissue	Mansour et al. [35]	- 10 urines matched with 14 tissues from cancer patients - Catheterized urine vs. cancer tissue - Comparisons between age groups and genders - Increased alpha diversities in male tissues	- Different age group, gender, sample type showed different microbial abundances - Different tissues from same patient showed almost same microbial compositions - Microbiome between urine and tissue are shared
	Pederzoli et al. [36]	- Matched midstream urines, tumorous tissues, and non-tumorous tissues from 21 men and 8 men - Midstream urines from 20 patients and 59 healthy controls	- Significant differences in microbial abundances between patients’ urines and healthy controls’ urines - Tumorous tissues showed more Burkholderia - More than 80% of the bacterial families were shared between urine and tissue microbiome

**Table 2 biomedicines-10-01783-t002:** Summary of beneficial microbes in bladder cancer.

Microbe	Author	Findings
*Bifidobacterium infantis* (BI)	Cinque et al. [63]	- BI can specifically induce apoptosis of anaerobic tumors.
	Tang et al. [64]	- Development of BI-TK/GCV. - BI-TK/GCV suicide inhibited tumor growth through increasing caspase-3 expression and inducing apoptosis.
	Yin et al. [65]	- BI-TK/GCV increased Fas, FasL, Cyt-C, and caspase-9 expression involving both extrinsic and intrinsic apoptosis.
	Jiang et al. [66]	- BI-TK/GCV decreased expressions of some proteins (PCNA, PKM2, HKK-1, PFK-B, CD146, Prx-I) showing antitumor effect. - Downregulation of Prx-I is related to NF-kB pathway showing apoptotic effect on tumor.
*Bifidobacterium longum* (BL)	Kitagawa et al. [67]	- Development of a recombinant BL displaying WT1 protein - Oral cancer vaccine can boost the antitumor effect of anti-PD-L1 and even do it alone by boosting T cell immunity.
*Lactococcus lactis* (LL)	Asensi et al. [68]	- Oral immunization with LL-rSEB induced a protective immune response in infectious mouse model.
	Reis et al. [69]	- Intravesical LL-rSEB injection showed better balance between apoptosis and cell proliferation with decreased VEGF, HIF, NOX4, and MMP-9 proteins. - Intravesical LL-rSEB injection showed more effectiveness than intravesical BCG injection in bladder cancer rat model.
*Lactobacillus casei* strain Shirota (LcS)	Asano et al. [70]	- Subcutaneously MBT-2 implanted mouse model. - LcS feeding and intravenous injection induced tumor regression.
	Aso et al. [71]	- 138 patients with superficial bladder cancer. - Oral LcS intake decreased recurrence rates in patients with primary multiple tumors and recurrent single tumors.
	Naito et al. [72]	- 207 patients with superficial bladder cancer. - Combination of oral LcS with intravesical epirubicin showed lower recurrence rates.
	Matsuzaki [73]	- Multiple cell lines in mice and guinea pigs. - Intravenous, intralesional, subcutaneous, and intrapleural injection of LcS showed increased pro-inflammatory cytokines, T cell, and NK cell activities.
	Kato et al. [74]	- LcS induced secretion of IL-12 and IFN-γ in mouse splenocytes. - Increased IFN-γ production in spleen of LcS fed mouse.
	Takahashi et al. [75]	- Subcutaneously and orthotopically MBT-2 implanted mouse model. - LcS subcutaneous and intravesical injection showed higher tumor regression rates and lower tumor weights. - LcS intravesical injection showed higher IFN-γ and TNF-α transcription in bladder tissues.
	Hori et al. [76]	- Oral administration of LcS in aged mice. - LcS increased Nk cell activity in blood mononuclear cells and splenocytes.
	Shida et al. [77]	- Coculture of LcS with PBMC from healthy volunteers. - Monocytes are essential for IFN-γ secreting T cell activity and NK cell activity.
	Takeda et al. [78]	- Middle-aged healthy volunteers (30–45 yrs) with low NK cell activity before oral LcS intake showed significantly elevated NK cell activity in peripheral blood.
	Dong et al. [79]	- Elderly healthy volunteers (55–74 yrs) with oral LcS intake showed increased CD69^+^, CD25^+^ Th cells, CD8^-^ NK cells, and IL-10 to IL-12 ratio.
	Seifert et al. [80]	- Healthy males aged 18–60 yrs showed no difference in NK cell activity according to oral LcS intake.
*Lactobacillus rhamnosus* strain GG (LGG)	Lim et al. [81]	- Subcutaneously MB49 implanted mouse model. - LGG feeding showed decreased tumor sizes, increased T lymphocyte population in spleens, and increased lymphocyte population in tumors.
	Seow et al. [82]	- Similar to BCG, intravesical LGG injection increased function-related genes and NK cell population in healthy mouse bladder and iliac lymph nodes.
	Seow et al. [83]	- Using orthotopically MB49 implanted mouse model. - Intravesical LGG induced tumor regression, macrophage and neutrophil infiltration to tumor, and inhibited metastasis-related and tissue-remodelling enzymes.
	Cai et al. [84]	- Mice-derived dendritic cells, neutrophils, T lymphocytes. - T cell IFN-r and IL-2 secretion are related to LGG dose and exposure time. - Dendritic cells and neutrophil activations are involved in this immune-related process.

## Data Availability

Not applicable.

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
