# Peer review of "Bacteria for Treatment: Microbiome in Bladder Cancer"

_biomedicines, 2022, doi:10.3390/biomedicines10081783_

Round 1

Reviewer 1 Report

The review highlights the starting material for microbiome research in the field of bladder cancer and relevant microbes with therapeutic potential for bladder cancer treatment.

In the abstract, please add in human body sites, which would sound better than human bodies.

In the introduction state of art and references in concerns BC incidence rate worldwide by the latest Globocan Cancer Statistics,

the clinicalpathological bladder cancer disease types and recent prognostic factors and treatment options for MBICs and NMIBCs are missing.

By doing a basic Medline, there are many other review in this field, with more hexaustive content. 

(("microbiome s"[All Fields] OR "microbiomic"[All Fields] OR "microbiomics"[All Fields] OR "microbiota"[MeSH Terms] OR "microbiota"[All Fields] OR "microbiome"[All Fields] OR "microbiomes"[All Fields]) AND ("urinary bladder neoplasms"[MeSH Terms] OR ("urinary"[All Fields] AND "bladder"[All Fields] AND "neoplasms"[All Fields]) OR "urinary bladder neoplasms"[All Fields] OR ("bladder"[All Fields] AND "cancer"[All Fields]) OR "bladder cancer"[All Fields])) AND (review[Filter])

At least (n= 9/n = 34 total reviews) focus on microbiome involvement and how to study its composition and diversity in bladder cancer with recent findings and insightful comments.

Please improve citations. For instance, the human microbiome in genitourinary cancers DOI: 10.21037/atm-20-2976; NMIBCs, immunity and microbial signature relationships could be find by citing DOI 10.3389/fchem.2020.00600

The review aim should be better stated  by clarifying that the advances in immunotherapy with the great impact especially on BC, since the innate and adaptive immune systems play a critical role in BCG immunotherapy response.

Line 42-44 references suppporting the sentences are necessary.

Minor points:

Line 41 please use human body sites rather than human bodies.

Please, check line 76-80 there is a ripetition.

2. Methods for microbiome study in bladder cancer

Line 82, Kraken2 and MetaPh1An3 are you sure that both are the only tools available for Shotgun sequencing ?

3.1 Research using urine

Line 153: Catheterized urine from voided urine is different, too. Where are references supporting the sentence ?

3.2 Research using tissue

Section incomplete and line 187-192 to remodulate. The TCGA has standardized pipeline for molecular tissue and blood (cancer/normal) profiling 

and it is true that tissue-embedded microbes remained an outstanding challenge for use of these dataset.

However, the whole body of literature in concerns the human tumor microbiome and microbial signatures within tumor tissue is missing, as well as The Cancer Microbiome analysis (TCMA).

The work by Rodriguez et al. and Li et al. adopt WES and WTS both by using the TCGA and PathoScope workflow respectively.

3.3 Research using urine and tissue and 4. concluding remarks

The reviewed studies are a list of findings without insightful interpretation.

Author Response

The authors would like to thank the reviewer for providing detailed comments. We have tried to answer each of these comments sincerely. Modifications of the contents are highlighted in yellow, and description comments have been added. As suggested by the reviewers, all the words and grammar have been thoroughly corrected. 

The review highlights the starting material for microbiome research in the field of bladder cancer and relevant microbes with therapeutic potential for bladder cancer treatment.

In the abstract, please add in human body sites, which would sound better than human bodies.

 --> Thank you for this suggestion. We have modified the sentence “There are many kinds of microbes residing in human bodies.” to “The human body contains a variety of microbes.” Additionally, thorough proofreading and grammar checking was performed.

In the introduction state of art and references in concerns BC incidence rate worldwide by the latest Globocan Cancer Statistics, the clinicalpathological bladder cancer disease types and recent prognostic factors and treatment options for MBICs and NMIBCs are missing.

 --> We have added a description providing an overview of the clinicopathological aspects of bladder cancer.

By doing a basic Medline, there are many other review in this field, with more hexaustive content.

(("microbiome s"[All Fields] OR "microbiomic"[All Fields] OR "microbiomics"[All Fields] OR "microbiota"[MeSH Terms] OR "microbiota"[All Fields] OR "microbiome"[All Fields] OR "microbiomes"[All Fields]) AND ("urinary bladder neoplasms"[MeSH Terms] OR ("urinary"[All Fields] AND "bladder"[All Fields] AND "neoplasms"[All Fields]) OR "urinary bladder neoplasms"[All Fields] OR ("bladder"[All Fields] AND "cancer"[All Fields]) OR "bladder cancer"[All Fields])) AND (review[Filter])

At least (n= 9/n = 34 total reviews) focus on microbiome involvement and how to study its composition and diversity in bladder cancer with recent findings and insightful comments.

Please improve citations. For instance, the human microbiome in genitourinary cancers DOI: 10.21037/atm-20-2976; NMIBCs, immunity and microbial signature relationships could be find by citing DOI 10.3389/fchem.2020.00600

--> The review articles have been checked and cited. Through this process, we have made the following changes: "Increasing evidence indicates that the microbiome influences the development and treatment of genitourinary malignancies [23-26]." and "An understanding of how the bladder microbiota affects immunotherapy response, leading to patients who are not responsive to BCG treatment is necessary [27]."

The review aim should be better stated by clarifying that the advances in immunotherapy with the great impact especially on BC, since the innate and adaptive immune systems play a critical role in BCG immunotherapy response.

 --> We have provided a detailed description of the immunotherapeutic drugs that have been approved and are undergoing trials. A special focus was placed on nadofaragene firadenovec and enfortumab vedotin, in accordance with our view of the therapeutic effects of microbiomes.

Line 42-44 references suppporting the sentences are necessary.

--> There was no citation in lines 42-44, which stated that "Fecal microbiota transplantation (FMT), diet and prebiotics, symbiotic microbiota consortia, engineered symbiotic bacteria, and microbiota-derived proteins and metabolites are studied for microbiome-based therapeutics." A citation was thus inserted here.

Minor points:

1. Line 41 please use human body sites rather than human bodies.

Please, check line 76-80 there is a ripetition.

  --> Line 41 stated that “With advances in sequencing and bioinformatics analyses, we now know that so many kinds of microbes reside in our human bodies, even in the urinary tract known to be sterile”. The sentence has been changed to “Using bioinformatics and sequencing technologies, many different microbes have been found to live in various sites of the human body, including the urinary tract, which is commonly believed to be sterile” changing the term “our human bodies” to “various sites of the human body”. Lines 76-80 were removed as they were repeated sentences.

2. Methods for microbiome study in bladder cancer

Line 82, Kraken2 and MetaPh1An3 are you sure that both are the only tools available for Shotgun sequencing ?

 --> Line 82 stated that “Kraken2 and MetaPhlAn3 are the tools for shotgun metagenomic sequencing”. We did not intend to convey that Kraken2 and MetaPhlAn3 are the only tools for shotgun sequencing. Therefore, the sentence has been changed to “Kraken2 and MetaPhlAn3 are useful tools for shotgun metagenomic sequencing”.

3.1 Research using urine

Line 153: Catheterized urine from voided urine is different, too. Where are references supporting the sentence ?

 --> This sentence in the manuscript was derived from the studies of Hourigan et al. and Oresta et al. However, the contents were moved to the top because of their ineffective visibility. The link between the sentence and the citations was indicated.

3.2 Research using tissue

Section incomplete and line 187-192 to remodulate. The TCGA has standardized pipeline for molecular tissue and blood (cancer/normal) profiling and it is true that tissue-embedded microbes remained an outstanding challenge for use of these dataset.

However, the whole body of literature in concerns the human tumor microbiome and microbial signatures within tumor tissue is missing, as well as The Cancer Microbiome analysis (TCMA).

The work by Rodriguez et al. and Li et al. adopt WES and WTS both by using the TCGA and PathoScope workflow respectively.

--> The concept and meaning of TCMA, as well as the statement that Pathoscope 2.0 was used for tissue analysis and added to the revised paper. Line 187-192, the summary of the section, has been modified to include a statement regarding the limitations of TCGA and the benefits of TCMA.

3.3 Research using urine and tissue and 4. concluding remarks

The reviewed studies are a list of findings without insightful interpretation.

--> In the research using urine and tissue section, we suggested shotgun metagenomic sequencing to identify the microbes more accurately. In the concluding remarks, we have suggested future directions for microbiome studies based on other review articles by emphasizing the potential of diverse microbes, the importance of multi-omics research including metabolomics, as well as the use of microbial signatures in prediction algorithms.

Reviewer 2 Report

In this manuscript titled “Bacteria for treatment: Microbiome in bladder cancer”, the authors reviewed the recent discoveries of microbiome in the bladder, especially in the bladder with cancer. Furthermore, the authors discussed the current efforts of using specific microbes identified in the human bladder as potential treatments for bladder cancer. Overall, this is a timely and interesting review. It is well written. I only have several minor comments.

Minor issues:

1. In the section “2. Methods for microbiome study in bladder cancer”, the authors reviewed many aspects of the methods, however, the authors didn’t review the different methods for collecting urine. Different collection methods have different possibility for microbe contamination, therefore, it is important to discuss the pros and cons of those different urine collection methods.

2. From line 76 to line 87, the authors mentioned different computational processes and analyses of microbiome study, however, the authors didn’t introduce each method at all. For people who are not from bioinformatics background, it is hard to understand the differences of these computational methods. Therefore, it may be interesting to briefly cover the underlying principle of each computational model mentioned here and to summarize the differences between different computational methods.

3. Line 159-164, the authors mentioned “Liu et al. collected 22 carcinoma tissues and 12 adjacent normal 160 tissues”. How did researchers distinguish carcinoma tissue vs normal tissue? The standard used is very critical because it may affect the microbiome detected. Please briefly describe the standard used to identify cancer tissue vs normal tissue in different studies.

4. In the section “3). Research using both urine and tissue”, the authors mentioned tissues were collected from cancer patients and healthy controls, but the authors didn’t explicitly state whether the tissues collected from cancer patients were cancer tissues or tissues not necessarily in the cancer site (kind of normal tissue). Please be more accurate in the description.

5. Grammar needs to be checked throughout the manuscript. For example, line 32, “multiple” is not an accurate word here; line 42, “known to be sterile” should be “previously known to be sterile”.

Author Response

In this manuscript titled “Bacteria for treatment: Microbiome in bladder cancer”, the authors reviewed the recent discoveries of microbiome in the bladder, especially in the bladder with cancer. Furthermore, the authors discussed the current efforts of using specific microbes identified in the human bladder as potential treatments for bladder cancer. Overall, this is a timely and interesting review. It is well written. I only have several minor comments.

 --> The authors would like to thank the reviewer for providing detailed comments. We have tried to answer each of these comments sincerely. Modifications of the contents are highlighted in yellow, and description comments have been added. As suggested by the reviewers, all the words and grammar have been thoroughly corrected. 

Minor issues:

  1. In the section “2. Methods for microbiome study in bladder cancer”, the authors reviewed many aspects of the methods, however, the authors didn’t review the different methods for collecting urine. Different collection methods have different possibility for microbe contamination, therefore, it is important to discuss the pros and cons of those different urine collection methods.

 --> Thank you for this comment. Lines 112–120 describe the methods used to collect urine. These contents were originally located in the section below but were moved here to increase their visibility.

  1. From line 76 to line 87, the authors mentioned different computational processes and analyses of microbiome study, however, the authors didn’t introduce each method at all. For people who are not from bioinformatics background, it is hard to understand the differences of these computational methods. Therefore, it may be interesting to briefly cover the underlying principle of each computational model mentioned here and to summarize the differences between different computational methods.

  --> In lines 136–156, the computational tools have been described in greater detail. In addition, we describe insights from our experience regarding computational tools.

  1. Line 159-164, the authors mentioned “Liu et al. collected 22 carcinoma tissues and 12 adjacent normal 160 tissues”. How did researchers distinguish carcinoma tissue vs normal tissue? The standard used is very critical because it may affect the microbiome detected. Please briefly describe the standard used to identify cancer tissue vs normal tissue in different studies.

 --> For clarification, we have added the following: “During the surgical procedure, bladder carcinoma tissue samples were taken from the mucosal surface of the bladder, and normal tissues were located approximately five centimeters away from the cancerous tissues.” 

  1. In the section “3). Research using both urine and tissue”, the authors mentioned tissues were collected from cancer patients and healthy controls, but the authors didn’t explicitly state whether the tissues collected from cancer patients were cancer tissues or tissues not necessarily in the cancer site (kind of normal tissue). Please be more accurate in the description.

 --> We described the samples accurately so that they could be identified. Yellow indicates descriptions that have been newly added. 

  1. Grammar needs to be checked throughout the manuscript. For example, line 32, “multiple” is not an accurate word here; line 42, “known to be sterile” should be “previously known to be sterile”.

 --> According to the reviewers' comments, all the words and grammar have been thoroughly corrected. Line 32 stated that “When the tumor recurs or multiple, urologists should consider intravesical BCG instillation or radical cystectomy”. We have revised lines 52–56 to state that "However, urologists should consider intravesical therapy if the tumor progresses to intermediate or high risk. BCG or radical cystectomy should be considered for high-risk, BCG-naive patients". And we have changed "known to be sterile" in line 42 to "which is commonly believed to be sterile" in line 87.

Round 2

Reviewer 1 Report

The authors have improved the revised review, most of my concerns have been addressed, one reference is missing, please add where previously suggested the relationships between NMIBCs, immunity and microbial signature by DOI 10.3389/fchem.2020.00600 and check for some typos. 

For instance, bacterial nomenclature italics at genus and species level all over the revised review and especially in the supplementary tables.

Author Response

The authors have improved the revised review, most of my concerns have been addressed, one reference is missing, please add where previously suggested the relationships between NMIBCs, immunity and microbial signature by DOI 10.3389/fchem.2020.00600 and check for some typos. 

For instance, bacterial nomenclature italics at genus and species level all over the revised review and especially in the supplementary tables.

Thank you for your comments.

  1. Using the article as a reference, ae emphasized the relationship between the NMIBC, immunity, and the microbiome by emphasizing the possible role of the microbiome in predicting responses to BCG.
  2. We skipped to fix it in italics. We modified the nomenclatures to italics on genus and species levels.